# APOLLO: AN ADAPTIVE PARAMETER-WISE DIAGONAL QUASI-NEWTON METHOD FOR NONCONVEX STOCHASTIC OPTIMIZATION

## ABSTRACT

In this paper, we introduce APOLLO, a quasi-Newton method for nonconvex stochastic optimization, which dynamically incorporates the curvature of the loss function by approximating the Hessian via a diagonal matrix. Importantly, the update and storage of the diagonal approximation of Hessian is as efficient as adaptive first-order optimization methods with linear complexity for both time and memory. To handle nonconvexity, we replace the Hessian with its rectified absolute value, which is guaranteed to be positive-definite. Experiments on three tasks of vision and language show that APOLLO achieves significant improvements over other stochastic optimization methods, including SGD and variants of Adam, in terms of both convergence speed and generalization performance. The implementation of the algorithm is available at *anonymous link*.

## 1 INTRODUCTION

Nonconvex stochastic optimization is of core practical importance in many fields of machine learning, in particular for training deep neural networks (DNNs). First-order gradient-based optimization algorithms, conceptually attractive due to their linear efficiency on both the time and memory complexity, have led to tremendous progress and impressive successes. A number of advanced first-order algorithms have emerged over the years to pursue fast and stable convergence, among which stochastic gradient descent (SGD) (Robbins & Monro, 1951; LeCun et al., 1998), equipped with momentum (Rumelhart et al., 1985; Qian, 1999; Bottou & Bousquet, 2008), has stood out for its simplicity and effectiveness across a wide range of applications (Hinton & Salakhutdinov, 2006; Hinton et al., 2012; Graves, 2013). However, one disadvantage of SGD is that the gradients in different directions are scaled uniformly, resulting in limited convergence speed and sensitive choice of learning rate, and thus has spawned a lot of recent interests in accelerating SGD from the algorithmic and practical perspectives.

Recently, many *adaptive* first-order optimization methods have been proposed to achieve rapid training progress with element-wise scaled learning rates, and we can only mention a few here due to space limits. In their pioneering work, Duchi et al. (2011) proposed AdaGrad, which scales the gradient by the square root of the accumulative square gradients from the first iteration. While AdaGrad works well for sparse settings, its performance significantly degrades for dense settings, primarily due to the monotonic increase of the accumulation. Subsequently, several methods have been proposed with the intuition to limit the accumulation to a small window of past iterations, and in particular exponentially reduce the weight of earlier iterations. Notable works incorporating this method are RMSProp (Tieleman & Hinton, 2012), AdaDelta (Zeiler, 2012), and Adam (Kingma & Ba, 2015), among which Adam has become the default optimization algorithm across many deep learning applications because of its fast convergence speed and relatively consistent selections of hyper-parameters (Ruder, 2016; Zhang et al., 2020). However, it has been observed that these adaptive optimization methods may converge to bad/suspicious local optima, resulting in worse generalization ability than their non-adaptive counterparts (Wilson et al., 2017), or fail to converge due to unstable and extreme learning rates (Luo et al., 2019).

Quasi-Newton methods have been widely used in solving convex optimization problems, due to their efficient computation and fast convergence rate (Broyden, 1967; Dennis & Moré, 1977). However, the stochastic, high-dimensional and nonconvex nature of many machine learning tasks, such as

training deep neural networks, has rendered many classical quasi-Newton methods ineffective and/or inefficient (Keskar & Berahas, 2016; Wang et al., 2017; Yao et al., 2020). Indeed, in many natural language processing (NLP) and computer vision (CV) tasks (He et al., 2016; Ma & Hovy, 2016; Luo et al., 2019), SGD (with momentum) is chosen as the optimizer, benefiting from its stable and efficient training and outstanding generalization.

In this work, we develop APOLLO, a quasi-Newton method for nonconvex stochastic optimization to simultaneously tackle the aforementioned challenges of stochastic variance, nonconvexity and inefficiency. Algorithmically, APOLLO dynamically incorporates the curvature of the objective function with diagonally approximated Hessian. It only requires first-order gradients and updates the approximation of the Hessian diagonally so that it satisfies a parameter-wise version of the weak secant condition (Wolfe, 1959). To handle nonconvexity, we replace the Hessian with its rectified absolute value, the computation of which is also efficient under our diagonal approximation, yielding an efficient optimization algorithm with linear complexity for both time and memory (§3). Experimentally, through three tasks on CV and NLP with popular deep neural networks, including ResNets (He et al., 2016), LSTMs (Hochreiter & Schmidhuber, 1997) and Transformers (Vaswani et al., 2017), we demonstrate that APOLLO significantly outperforms SGD and variants of Adam, in terms of both convergence speed and generalization performance (§4).

## 2 BACKGROUNDS

In this section, we set up the notations on nonconvex stochastic optimization, briefly review the (quasi-) Newton methods, and discuss the problems of applying quasi-Newton methods to nonconvex stochastic optimization that we attempt to study in the rest of the paper.

### 2.1 NONCONVEX STOCHASTIC OPTIMIZATION

In this paper, we consider the following nonconvex stochastic optimization problem:

$$\min_{\theta \in \mathcal{R}^d} f(\theta) = \mathbb{E}[l(\theta; \Gamma)] \tag{1}$$

where $l : \mathcal{R}^d \times \mathcal{R}^n \to \mathcal{R}$ is a continuously differentiable (and possible nonconvex) function, $\theta \in \mathcal{R}^d$ denotes the parameter to be optimized, $\Gamma \in \mathcal{R}^n$ denotes a random variable with distribution function P, and $\mathbb{E}[\cdot]$ denotes the expectation w.r.t $\Gamma$. Intuitively, $\Gamma$ incorporates noises in $f$, leading to a stochastic objective function. A special case of (1) that arises frequently in machine learning is the empirical risk minimization problem:

$$\min_{\theta \in \mathcal{R}^d} f(\theta) = \frac{1}{N} \sum_{i=1}^{N} l_i(\theta) \tag{2}$$

where $l_i : \mathcal{R}^d \to \mathcal{R}$ is the loss function corresponding to the $i$-th data, and $N$ is the number of data samples that is assumed to be extremely large. Objective functions may also have other sources of noise than data subsampling, such as dropout (Srivastava et al., 2014) in deep neural networks.

**Decoupled Parameters.** In this work, we consider a setting of decoupled parameters: $\theta = \{\theta^{(l)}, l = 1, \ldots, L\}$. Intuitively, under this setting the parameter $\theta$ is decoupled into a sequence of parameters serving different functionalities. For example, in neural network training the parameters of a neural network can be naturally decoupled into the parameters of different layers or modules.

### 2.2 NEWTON AND QUASI-NEWTON METHODS

Newton's method usually employs the following updates to solve (1):

$$\theta_{t+1} = \theta_t - H_t^{-1} g_t \tag{3}$$

where $g_t = \nabla f(\theta_t)$ is the gradient at $\theta_t$ and $H_t = \nabla^2 f(\theta_t)$ is the Hessian matrix. The convergence rate of Newton's method is *quadratic* under standard assumptions (Nocedal & Wright, 2006). However, major challenges with this method are i) the expensive computation of the inverse Hessian at every iteration and the corresponding quadratic memory complexity; and ii) the limitation to convex functions (nonconvexity results in negative curvature of $H_t$ and misleads the update directions).

A standard alternative to Newton's method is a class of quasi-Newton methods, which have been widely used in solving convex deterministic optimization problem:

$$\theta_{t+1} = \theta_t - \eta_t B_t^{-1} g_t \tag{4}$$

where $\eta_t$ is the stepsize (a.k.a learning rate), $B_t$ is an approximation to the Hessian matrix $\nabla^2 f(\theta_t)$ at $\theta_t$, which is updated based on the well-known secant equation:

$$B_{t+1} = \underset{B}{\arg\min} \|B - B_t\|$$
$$\text{s.t.} \quad B_{t+1} s_t = y_t \quad \text{(secant equation)} \tag{5}$$

where $s_t = \theta_{t+1} - \theta_t$ and $y_t = g_{t+1} - g_t$. $B_{t+1}$ is, in the sense of some matrix norm, the closest to $B_t$ among all symmetric matrices that satisfy the secant equation. Each choice of the matrix norm results in a different update formula, such as DFP (Davidon, 1991; Fletcher, 1987) and BFGS (Broyden, 1970; Fletcher, 1970; Goldfarb, 1970; Shanno, 1970). The popularity of this method is due to the fact that only the gradient of the objective function is required at each iteration. Since no second derivatives (Hessian) are required, quasi-Newton methods are sometimes more efficient than Newton's method, especially when the computation of Hessian is expensive. To further reduce memory cost, one seminal work is the limited memory BFGS (L-BFGS) (Liu & Nocedal, 1989; Byrd et al., 1995) that achieves desirable linear computational and memory complexity by approximating the Hessian as a series of sum of first order information from previous iterations.

### 2.3 PROBLEMS OF QUASI-NEWTON METHODS

Despite their impressive successes on convex deterministic optimization, quasi-Newton methods suffer from their own problems in more challenging scenarios. In this section, we mainly discuss three problems preventing quasi-Newton methods from being applied to the scenario of large-scale nonconvex stochastic optimization. Due to these problems, no quasi-Newton methods (to our best knowledge) designed for nonconvex optimization consistently outperform adaptive first-order algorithms w.r.t convergence speed and generalization performance. The main goal of this work is to algorithmically design and experimentally demonstrate a novel quasi-Newton method, in hope of improving the convergence speed and generalization performance of nonconvex stochastic optimization eventually.

**Stochastic Variance.** One challenge of quasi-Newton methods on nonconvex stochastic optimization (1) is the variance introduced by the stochastic nature of the problem. At each iteration, only the stochastic gradient $g_t$ is available, which is an unbiased estimation of the gradient $\nabla f(\theta_t)$ and may lead to an erroneous approximation of Hessian (Byrd et al., 2011).

**Nonconvexity.** Another key challenge in designing such quasi-Newton methods lies in the difficulty of preserving the positive-definiteness of $B_t$ in (5), due to the nonconvexity of the objective function. What is worse is that performing line search is infeasible in the stochastic setting, due to the presence of noise in the stochastic gradients (Wang et al., 2017).

**Computational and Memory Efficiency.** Even though quasi-Newton methods are more efficient than Newton's method, the time and memory complexities are still relatively large compared with adaptive first-order methods. For instance, L-BFGS requires to store first-order information from $m$ previous iterations with commonly $m \geq 5$, which is still too expensive for deep neural networks containing millions of parameters. Moreover, adapting quasi-Newton methods to nonconvex stochastic optimization probably introduces additional computation, further slowing down these methods.

## 3 ADAPTIVE PARAMETER-WISE DIAGONAL QUASI-NEWTON

With the end goal of designing an efficient quasi-Newton method to solve the problem in (1) in mind, we first propose to approximate the Hessian with a diagonal matrix, whose elements are determined by the variational approach subject to the *parameter-wise* weak secant equation (§3.1). Then, we explain our stepsize bias correction technique to reduce the stochastic variance in §3.2. To handle nonconvexity, we directly use the rectified absolute value of the diagonally approximated Hessian as the preconditioning of the gradient (§3.3). The initialization technique of APOLLO allows us to eliminate one hyper-parameter (§3.4). At last, we provide a theoretical analysis of APOLLO's convergence in both convex optimization and nonconvex stochastic optimization (§3.5). The pseudo-code is shown in Algorithm 1.

### 3.1 QUASI-NEWTON METHODS WITH DIAGONAL HESSIAN APPROXIMATION

As discussed in Bordes et al. (2009), designing an efficient stochastic quasi-Newton algorithm involves a careful trade-off between the sparsity of the approximation matrix $B_t$ and the quality of

its approximation of the Hessian $H_t$, and diagonal approximation is a reasonable choice (Becker et al., 1988; Zhu et al., 1999). If $B$ is chosen to be a diagonal matrix satisfying (5), one can obtain a formula similar to the SGD-QN algorithm (Bordes et al., 2009).

An alternative of the secant equation in the updating formula (5), as first introduced by Nazareth (1995), is the weak secant equation (Dennis & Wolkowicz, 1993):

$$B_{t+1} = \underset{B}{\text{argmin}} \|B - B_t\|$$
$$\text{s.t.} \quad s_t^T B_{t+1} s_t = s_t^T y_t \quad \text{(weak secant equation)} \tag{6}$$

The motivation of using the weak secant condition in diagonal quasi-Newton method is straightforward: the standard mean-value theorem might not necessarily hold for vector-valued functions expressed in the secant equation, $B_{t+1} s_t = y_t \approx \nabla^2 f(\theta_t) s_t$. Thus, we do not know whether there exists a vector $\tilde{\theta} \in \mathcal{R}^d$ such that $y_t = \nabla^2 f(\tilde{\theta}) s_t$ (Dennis & Moré, 1977). On the other hand, the Taylor theorem ensures that there exists such $\tilde{\theta}$ that $s_t^T y_t = s_t^T \nabla^2 f(\tilde{\theta}) s_t$, leading to the reasonable assumption of the weak secant condition (6).

Based on the variational technique (Zhu et al., 1999), the solution of (6) with Frobenius norm is:

$$\Lambda \triangleq B_{t+1} - B_t = \frac{s_t^T y_t - s_t^T B_t s_t}{\|s_t\|_4^4} \text{Diag}(s_t^2) \tag{7}$$

where $s_t^2$ is the element-wise square vector of $s_t$, $\text{Diag}(s_t^2)$ is the diagonal matrix with diagonal elements from vector $s_t^2$, and $\|\cdot\|_4$ is the 4-norm of a vector.

**Parameter-Wise Weak Secant Condition.** However, in optimization problems with high-dimensional parameter space, such as training deep neural networks with millions of parameters, the weak secant condition might be too flexible to produce a good Hessian approximation. In the setting of decoupled parameters (§2.1), we propose a parameter-wise version of the weak secant equation to achieve a trade-off between the secant and weak secant conditions: for each parameter $\theta^{(l)} \in \theta$, we update $B$ corresponding to $\theta^{(l)}$ by solving (6) individually. Remarkably, the secant condition restricts $B$ with an equation of a $d$-dimensional vector, while the weak secant condition relaxes it with a 1-dimensional scalar. The parameter-wise weak secant condition expresses the restriction as a $l$-dimension vector ($1 < l < d$), resulting in a reasonable trade-off. The updating formula is the same as (7) for each parameter-wise $B$.

## 3.2 STEPSIZE BIAS CORRECTION

To mitigate the stochastic variance problem in stochastic quasi-Newton methods, APOLLO utilizes stepsize bias correction on the stochastic gradients at each step $t$. We know that the optimal stepsize $\eta_t$ equals to 1 w.r.t the *quadratic approximation* underlying Newton's method, if the Hessian approximation $B_t$ and the stochastic gradient $g_t$ are close to the exact Hessian $H_t$ and gradient $\nabla f(\theta_t)$, respectively. Inspired by this, we correct the stepsize bias in the stochastic gradient $g_t$ by replacing it with a corrected gradient $g_t' = \eta_t g_t$. Together with the corresponding corrected $y_t' = g_{t+1}' - g_t' = \eta_t y_t$, we correct the updating term $\Lambda$ of $B_t$ in (7) by replacing $y_t$ with $y_t'$:

$$\Lambda' = \frac{s_t^T y_t' - s_t^T B_t s_t}{\|s_t\|_4^4} \text{Diag}(s_t^2) = -\frac{d_t^T y_t + d_t^T B_t d_t}{\|d_t\|_4^4} \text{Diag}(d_t^2) \tag{8}$$

where $d_t = -s_t/\eta_t = B_t^{-1} g_t$ is the corrected update direction. Note that after applying the step bias correction, the update formula of $B_t$ in (8) is independent with the stepsize $\eta_t$, eliminating the stepsize bias. Technically, the stepsize bias correction is designed to reduce the stochastic variance, rather than entirely discarding the stepsize. The APOLLO algorithm (Algorithm 1) still incorporates the stepsize at every iteration to enforce convergence.

Based on previous studies, incorporating exponential moving averages (EMVs) for the stochastic gradients significantly reduces the variance (Kingma & Ba, 2015). We follow these works and apply EMV to $g_t$, together with the initialization bias correction:

$$m_{t+1} = \frac{\beta(1 - \beta^t)}{1 - \beta^{t+1}} m_t + \frac{1 - \beta}{1 - \beta^{t+1}} g_{t+1} \tag{9}$$

where $0 < \beta < 1$ is the decay rate of EMV and $y_t$ in (8) is written as $m_{t+1} - m_t$. Note that we do not apply moving average methods to the approximated Hessian, though the diagonal matrix is easier to be explicitly formed to average than full matrices. Investigating the moving average of the diagonal $B_t$ might be an interesting direction of future work.

**Algorithm 1:** APOLLO, our proposed algorithm for nonconvex stochastic optimization. All operations on vectors are element-wise. Good default settings are $\beta = 0.9$ and $\epsilon = 1e^{-4}$.

**Initial:** $m_0, d_0, B_0 \leftarrow 0, 0, 0$         `// Initialize` $m_0, d_0, B_0$ `to zero`
**while** $t \in \{0, \ldots, T\}$ **do**
     **for** $\theta \in \{\theta^1, \ldots, \theta^L\}$ **do**
         $g_{t+1} \leftarrow \nabla f_t(\theta_t)$          `// Calculate gradient at step` $t$
         $m_{t+1} \leftarrow \frac{\beta(1-\beta^t)}{1-\beta^{t+1}} m_t + \frac{1-\beta}{1-\beta^{t+1}} g_{t+1}$    `// Update bias-corrected moving`
         average
         $\alpha \leftarrow \frac{d_t^T(m_{t+1}-m_t)+d_t^T B_t d_t}{(\|d_t\|_4+\epsilon)^4}$      `// Calculate coefficient of` $B$ `update`
         $B_{t+1} \leftarrow B_t - \alpha \cdot \text{Diag}(d_t^2)$          `// Update diagonal Hessian`
         $D_{t+1} \leftarrow \text{rectify}(B_{t+1}, 0.01)$          `// Handle nonconvexity`
         $d_{t+1} \leftarrow D_{t+1}^{-1} m_{t+1}$          `// Calculate update direction`
         $\theta_{t+1} \leftarrow \theta_t - \eta_{t+1} d_{t+1}$          `// Update parameters`
     **end**
**end**

### 3.3 RECTIFIED ABSOLUTE VALUE OF HESSIAN FOR NONCONVEXITY

To guarantee convergence, quasi-Newton methods require the approximated Hessian matrix $B_t$ to be positive definite at each step. The common strategy in previous studies is to solve the updating formula in (5) by restricting the candidate matrix $B$ to be symmetric positive definite. It is known that the BFGS update preserves the positive-definiteness of $B_{t+1}$ as long as the curvature condition $s_t^T y_t > 0$ holds, which can be guaranteed for strongly convex problem. For nonconvex problem, the curvature condition can be satisfied by performing a line search, which is, however, expensive or even infeasible in stochastic setting, because the exact function values and gradient information are unavailable. Wang et al. (2017) proposed the stochastic damped L-BFGS (SdLBFGS) method that implicitly generates a positive definite matrix without line search. However, it usually requires large history size ($m \geq 100$) to guarantee convergence, which is infeasible for large-scale optimization.

To handle nonconvexity, we adopt a different strategy that does not require the solution of $B_t$ in (5) to be positive definite. Intuitively, we search for $B_t$ that is a good approximation of the real Hessian, which is not necessarily positive definite in nonconvex problem. When we use $B_t$ as preconditioning to calculate the update direction, we use its absolute value: $|B_t| = \sqrt{B_t^T B_t}$, where $\sqrt{\cdot}$ is the positive definite square root of a matrix. The motivation of absolute value is straight-forward: for dimensions with large absolute values of curvature, the objective function could be very sharp and we would prefer to take relatively smaller steps than those flatter dimensions. Since APOLLO formulate $B_t$ as a diagonal matrix, the cost of computing $|B_t|$ is marginal.

**Rectified Absolute Value of** $B_t$    For nonconvex objective functions, there exist inflection points whose curvatures are zero. To prevent the steps from becoming arbitrarily large, we rectify the absolute value of $B_t$ with a convexity hyper-parameter $\sigma$:

$$D_t = \text{rectify}(B_t, \sigma) = \max(|B_t|, \sigma) \tag{10}$$

where the $\text{rectify}(\cdot, \sigma)$ function is similar to the rectified linear unit (ReLU) (Nair & Hinton, 2010) with threshold set to $\sigma$. The update direction in (8) is then $d_t = D_t^{-1} m_t$.

AdaHessian (Yao et al., 2020) used an idea similar to the absolute values of $B_t$ to handle nonconvexity, where the root mean square averaging is applied to compute the Hessian diagonal. Different from APOLLO, AdaHessian requires second-order information to compute the Hessian matvec oracle and approximate the Hessian diagonal using Hutchinson's method, which is significantly more costly.

### 3.4 INITIALIZATION

The rectified $D_t$ in (10) introduces one more hyper-parameter $\sigma$, limiting the application of APOLLO in practice. In this section, we show that the zero initialization approach in APOLLO, which initializes the moving average of gradient $m_0$, the parameter update direction $d_0$ and the diagonal approximation of Hessian $B_0$ as (vector of) zeros, leads to coupled stepsize $\eta$ and convexity $\sigma$, allowing us to eliminate one hyper-parameter of $\eta$ or $\sigma$.

**Coupled Stepsize $\eta$ and Convexity $\sigma$.** With the zero initialization of $m_0$, $d_0$ and $B_0$, the following theorem illustrates the relation between $\eta$ and $\sigma$ (details in Appendix A):

**Theorem 1.** *Given zero initialization of $m_0$, $d_0$, and $B_0$ and a fixed parameter intialization $\theta_0$. Suppose that we have two sets of hyper-parameters $\eta, \sigma$ and $\eta', \sigma'$ with the same ratio: $\frac{\eta}{\sigma} = \frac{\eta'}{\sigma'}$. Then the convergence trajectories of these two sets of hyper-parameters are exactly the same:*

$$\theta_t = \theta'_t, \ \forall t \in \{1, \ldots, T\}. \tag{11}$$

*where $\theta_t$ and $\theta'_t$ are the parameters of $(\eta, \sigma)$ and $(\eta', \sigma')$ at iteration $t$, respectively.*

From Theorem 1, we observe that $\eta$ and $\sigma$ are coupled with each other and in practice we only need to tune one of them, leaving the other fixed. Therefore, in our experiments (§4), we fix $\sigma = 0.01$ and tune $\eta$ on different problems[1].

**Learning Rate Warmup for APOLLO** As discussed in Kingma & Ba (2015), zero initialization leads to estimations biased towards zero in the initial iterations. For the moving average $m_t$, this bias can be corrected by dividing the bias-correction term (9). For $d_t$ and $B_t$, however, we cannot derive such bias correction terms. Fortunately, a simple linear warmup heuristic of $\eta$ at the beginning iterations achieves remarkably stable training.

## 3.5 CONVERGENCE ANALYSIS

Similar to previous work (Reddi et al., 2018; Chen et al., 2019; Zhuang et al., 2020), we omit the initialization bias correction step, i.e. we use $m_t = \beta_t m_{t-1} + (1 - \beta_t)g_t$, $0 < \beta_t < 1$, $\forall t \in [T]$.

We first analyze the convergence of APOLLO in convex optimization using the online learning framework (Zinkevich, 2003) for a sequence of convex cost functions $f_1(\theta), f_2(\theta), \ldots, f_T(\theta)$.

**Theorem 2.** *(Convergence in convex optimization) Let $\{\theta_t\}$ be the sequence from APOLLO. Suppose $\eta_t = \frac{\eta}{\sqrt{t}}$, $0 < \beta_t \leq \beta \leq 1 \|g_t\|_2 \leq G, \frac{\|D_{t-1}\|_1}{\eta_{t-1}} \leq \frac{\|D_t\|_1}{\eta_t}, \|\theta_t - \theta_{t'}\|_2 \leq D, \forall t, t' \in [T]$. For $\theta_t$ generated with the APOLLO algorithm, we have the following bound on the regret:*

$$R_T \leq \frac{\sqrt{T}D^2\|D_T\|_1}{2\eta(1-\beta)} + \frac{\eta G^2}{1-\beta}(2\sqrt{T} - 1) + \frac{D^2}{2(1-\beta)}\sum_{t=1}^{T}\frac{\beta_t^2}{\eta_t} \tag{12}$$

The following result falls as an immediate corollary of the above result.

**Corollary 2.1.** *Suppose $\beta_t = \beta\lambda^{t-1}, 0 < \lambda < 1$ in Theorem 2, we have*

$$R_T \leq \frac{\sqrt{T}D^2\|D_T\|_1}{2\eta(1-\beta)} + \frac{\eta G^2}{1-\beta}(2\sqrt{T} - 1) + \frac{D^2\beta^2}{2\eta(1-\beta)(1-\lambda^2)^2} \tag{13}$$

Theorem 2 implies the regret of APOLLO is upper bounded by $O(\sqrt{T})$. The conditions for Corollary 2.1, as in Reddi et al. (2018), can be relaxed to $\beta_t = \beta/t$ and still ensures a regret of $O(\sqrt{T})$.

For nonconvex case, we analyze the convergence rate of APOLLO with the similar derivations of that in Chen et al. (2019), since APOLLO belongs to the family of *generalized Adam-type* methods:

**Theorem 3.** *(Convergence in nonconvex stochastic optimization) Under the assumptions:*
- *$f$ is lower bounded and differentiable; $\|\nabla f(\theta) - \nabla f(\theta')\|_2 \leq L\|\theta - \theta'\|_2$, $\|D_t\|_\infty < L, \forall t, \theta, \theta'$.*
- *Both the true and stochastic gradient are bounded, i.e. $\|\nabla f(\theta_t)\|_2 \leq H$, $\|g_t\|_2 \leq H$, $\forall t$.*
- *Unbiased and independent noise in $g_t$, i.e. $g_t = \nabla f(\theta_t) + \zeta_t$, $\mathbb{E}[\zeta_t] = 0$, and $\zeta_i \perp \zeta_j$, $\forall i \neq j$.*

*Assume $\eta_t = \frac{\eta}{\sqrt{t}}$, $\beta_t \leq \beta \leq 1$ in non-increasing, $\frac{D_{t-1,j}}{\eta_{t-1}} \leq \frac{D_{t,j}}{\eta_t}$, $\forall t \in [T], j \in [d]$, then:*

$$\min_{t\in[T]} \mathbb{E}\left[\|\nabla f(\theta_t)\|_2^2\right] \leq \frac{L}{\sqrt{T}}(C_1\eta^2 H^2(1 + \log T) + C_2 d\eta + C_3 d\eta^2 + C_4) \tag{14}$$

where $C_1$, $C_2$, $C_3$ are constants independent of $d$ and $T$, $C_4$ is a constant independent of $T$, the expectation is taken w.r.t all the randomness corresponding to $\{g_t\}$. Theorem 3 implies the convergence rate for APOLLO in the non-convex case is $O(\log T/\sqrt{T})$, which is similar to Adam-type optimizer (Reddi et al., 2018; Chen et al., 2019). In addition, unlike Theorem 3.1 in Chen et al. (2019), Theorem 3 does not specify the bound of each update $\|\eta_t m_t/D_t\|_2$. This is because that, with conditions $\eta_t \leq \eta$, $\|g_t\|_2 \leq H$ and $D_t \geq 1$, it is straight-forward to derive the bound of $\|\eta_t m_t/D_t\|_2 \leq \eta H = G$.

---

[1]We changed $\sigma$ from 1 to 0.01 to make $\eta$ in a suitable range. See Appendix E.4 for details.

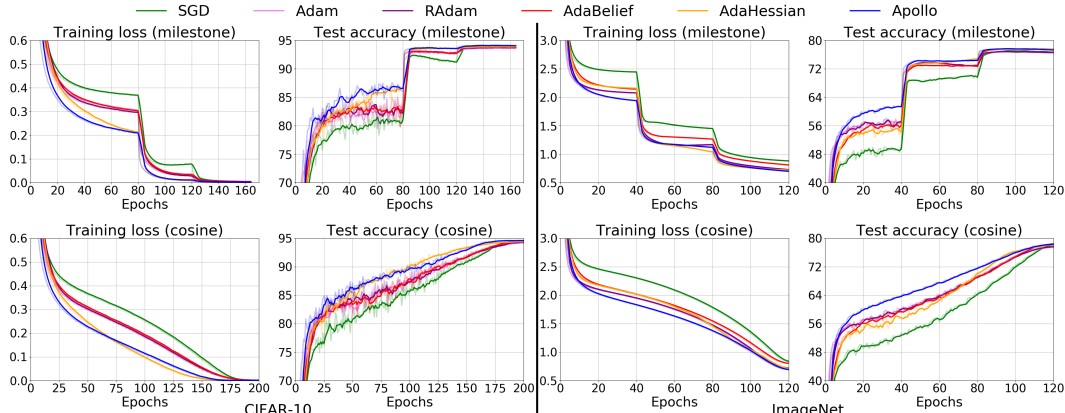

Figure 1: Training loss and test accuracy of ResNet-110 on CIFAR-10 and ResNeXt-50 on ImageNet, with two schedule strategies of learning rate decay.

## 4 EXPERIMENTS

To evaluate APOLLO, we conduct experiments on four benchmark datasets across three tasks of CV and NLP that are commonly used to evaluate optimization algorithms: CIFAR-10 (Krizhevsky & Hinton, 2009) and ImageNet (Deng et al., 2009) for image classification; One Billion Words (Chelba et al., 2013) for language modeling; and WMT 2014 English-German for neural machine translation. The five baseline methods we compare with are SGD with momentum (Bottou & Bousquet, 2008), Adam (Kingma & Ba, 2015), Rectified Adam (RAdam) (Liu et al., 2020), AdaBelief (Zhuang et al., 2020), and AdaHessian (Yao et al., 2020). Following Loshchilov & Hutter (2019), we decouple weight decays in Adam, RAdam, AdaBelief and AdaHessian in all the experiments[2]. For each experiment, we report the average over 5 runs. More detailed descriptions, results and analysis of the conducted experiments are provided in Appendix E.

### 4.1 IMAGE CLASSIFICATION

We begin our experiments with an evaluation of the convergence and generalization performance on image classification. We use ResNet-110[3] for CIFAR-10 and standard ResNeXt-50 (Xie et al., 2017) for ImageNet, respectively. The results on CIFAR-10 and ImageNet are presented in Figure 1 and Table 1, together with the five baselines. For each optimizer, we use two scheduling strategies of learning rate decay: i) milestone that decays the learning rate at the end of some predefined epochs; and ii) cosine annealing schedule proposed in Loshchilov & Hutter

Table 1: Test Acc. on CIFAR-10 and ImageNet.

| Method | CIFAR-10 | | ImageNet | |
|---|---|---|---|---|
| | milestone | cosine | milestone | cosine |
| SGD | 93.94 | 94.53 | 77.57 | 78.26 |
| Adam | 93.74 | 94.24 | 76.86 | 77.54 |
| RAdam | 93.88 | 94.38 | 76.91 | 77.68 |
| AdaBelief | 94.03 | 94.51 | 77.55 | 78.22 |
| AdaHessian | 93.97 | 94.48 | 77.61 | 78.02 |
| **APOLLO** | **94.21** | **94.64** | **77.85** | **78.45** |

(2017). All the optimization methods are comprehensively tuned, especially for the learning rate and the rate of weight decay. It is because that the strength of weight decay regularization is co-related with the learning rate, even though the decoupled weight decay technique (Loshchilov & Hutter, 2019) has been applied. The tuning information and the model details are provided in the Appendix E.1.

From Figure 1 and Table 1, we see that APOLLO outperforms the four first-order methods (SGD, Adam, RAdam and AdaBelief) on both the convergence speed and classification accuracy, demonstrating its effectiveness on training the ResNet architectures based on convolutional neural networks (CNNs) (LeCun et al., 1989). Comparing with AdaHessian, APOLLO obtains better test accuracy with similar convergence speed. Note that AdaHessian requires second-order information and is significantly more costly (detailed comparison of time and memory costs in Appendix F.3). Thus, we omit AdaHessian from the following experiments in the rest of this paper.

[2]For AdaBelief, we also tried standard $L_2$ regularization. But the accuracies are consistently worse than the models with decoupled weight decay.

[3]ResNet-110 is a modified (small) version of ResNet-18 to adapt the image size $32 \times 32$ in CIFAR-10.

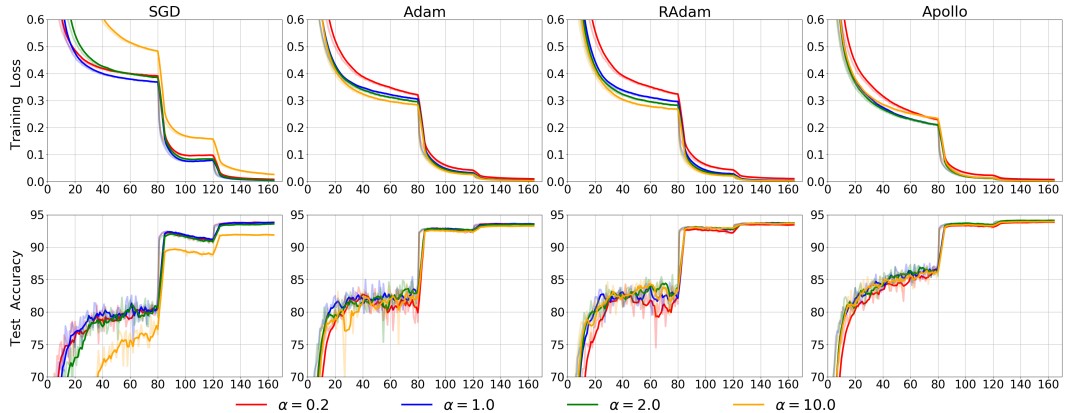

Figure 2: SGD, Adam, RAdam and APOLLO with different learning rates on CIFAR-10.

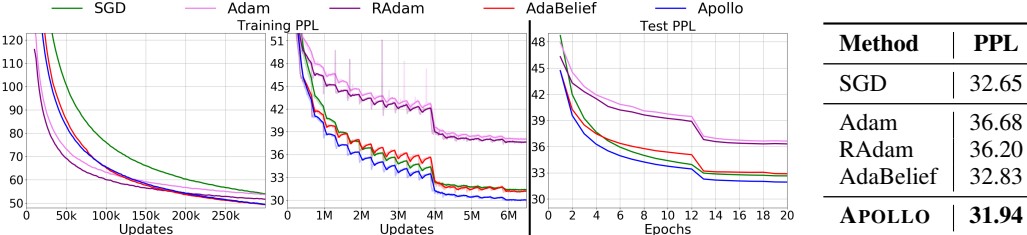

| Method | PPL |
|---|---|
| SGD | 32.65 |
| Adam | 36.68 |
| RAdam | 36.20 |
| AdaBelief | 32.83 |
| APOLLO | **31.94** |

Figure 3: Language modeling (LSTMs) on One Billion Words.    Table 2: Test PPL.

**Robustness to Learning Rate Change.**    Besides performance improvements, we also investigate the robustness of different optimization methods to the change of learning rate. For each optimizer, we use the learning rate in the previous experiment (Table 1) as the base, i.e. $0.1$ for SGD, $0.001$ for Adam and RAdam, and $0.01$ for APOLLO. Then, we explore different learning rates that are $\alpha$ times of the base learning rate, with $\alpha \in \{0.2, 1.0, 2.0, 10.0\}$. As mentioned above, we observed that the strength of weight decay regularization is co-related with the learning rate, even for Adam and RAdam with decoupled weight decay (Loshchilov & Hutter, 2019). To eliminate the impact of weight decay, we adjust the weight decay rates according to the factor $\alpha$. Experimental results with ResNet-110 on CIFAR-10 are summarized in Figure 2. After correcting the impact of weight decay, all the optimization methods, except SGD with $\alpha = 10.0$, achieve consistent model performance, while APOLLO slightly improves the robustness of model training over the three baseline methods.

## 4.2  LANGUAGE MODELING

To evaluate APOLLO on Recurrent Neural Networks (RNNs) that are applied in a wide range of NLP tasks (Graves, 2013), we conduct experiments on the One Billion Words dataset (Chelba et al., 2013), using a two-layer LSTM network for language modeling (details in Appendix E.2).

Figure 3 and Table 2 illustrate the perplexity (PPL) of training and test for APOLLO and four baseline methods, including SGD, Adam, RAdam and AdaBelief. As shown in Figure 3, although APOLLO is slower than Adam-type methods in the first few updates, its convergence is much faster after that. On generalization performance, APOLLO achieves significant improvements (more than 4.0 PPL points on test data) over Adam and RAdam. In addition, APOLLO also outperforms AdaBelief, which obtains the lowest PPL among the three Adam-type optimization methods[4]. This demonstrates the effectiveness of APOLLO on training LSTM-based neural architectures.

**Training Stability.**    From the middle plot of Figure 3 we see that the training losses of Adam and RAdam may suddenly increase. This occurred in all the runs of experiments using Adam and RAdam, and we selected these successfully converged — the loss went back to normal after some updates, and discarded those that failed to converge — the model crashed due to loss numerical overflow. The models optimized with APOLLO never suffered from this issue, demonstrating its stability.

---

[4]We found that AdaBelief is very sensitive to the value of $\epsilon$. The result in Table 2 is obtained using $\epsilon = 1e^{-12}$. With other values, e.g. $1e^{-8}$ or $1e^{-16}$, the PPL points of AdaBelief are even higher than Adam and RAdam. See Appendix E.2 for the details of hyper-parameter tuning.

### 4.3 NEURAL MACHINE TRANSLATION

To evaluate APOLLO on Attention-based Transformer architecture (Vaswani et al., 2017), we train the Transformer-base model on the WMT2014 English-German (EN-DE) dataset (around 4.5M sentence pairs). We use the same data preprocessing steps as in Ma et al. (2019) (details in Appendix E.3). We compare APOLLO with the same four baseline methods in the experiments of language modeling. For each experiment, we report the mean and standard variance over 5 runs. From Table 3, the first interesting observation is that SGD performs much worse than Adam-type methods (which is opposite to its behaviour for ResNet- and LSTM-based neural architectures). Similar observations about SGD were reported in (Yao et al., 2020; Zhang et al., 2020). Despite this, APOLLO obtains improvements over all the baseline methods for NMT with transformers.

Table 3: Test BLEU.

| Method | BLEU |
|---|---|
| SGD | 26.59±0.07 |
| Adam | 27.84±0.12 |
| RAdam | 28.15±0.15 |
| AdaBelief | 28.14±0.11 |
| **APOLLO** | **28.34**±0.10 |

## 5 RELATED WORK

**Stochastic Quasi-Newton Methods.** In the literature of (nonconvex) stochastic quasi-Newton methods, several algorithms have been developed recently for large-scale machine learning problems: oLBFGS (Schraudolph et al., 2007; Mokhtari & Ribeiro, 2015), RES (Mokhtari & Ribeiro, 2014), SFO (Sohl-Dickstein et al., 2014), SQN (Byrd et al., 2016), SdLBFGS (Wang et al., 2017), and AdaQN (Keskar & Berahas, 2016), among which only SdLBFGS and AdaQN are designed to solve nonconvex optimization problems. The SdLBFGS algorithm carefully controls the quality of modified BFGS updates to preserve the positive-definiteness of $B_t$ in (5) without line search. AdaQN shares a similar idea but is specifically designed for RNNs by refining the initial L-BFGS scaling, step acceptance control and choice of curvature information matrix, and adopting the SQN framework (Byrd et al., 2016). Different from these two methods, APOLLO does not require $B_t$ in (5) to be positive definite, but replacing it with its rectified absolute value to handle nonconvexity. Moreover, both SdLBFGS and AdaQN use the updating formula similar to L-BFGS and require even larger history size (commonly $\geq 100$) to guarantee convergence, preventing them from being applied to large-scale optimization. For comprehensive comparison of SdLBFGS with Apollo, we conducted experiments with small toy CNN models (details in Appendix G).

**Adaptive First-Order Methods.** From the diagonal approximation of Hessian, APOLLO is also related to those diagonally-scaled first-order algorithms, such as AdaGrad (Duchi et al., 2011), RMSProp (Tieleman & Hinton, 2012), AdaDelta (Zeiler, 2012), and Adam (Kingma & Ba, 2015). Subsequently, a number of techniques have emerged to theoretically justify and algorithmically improve Adam, including AMSGrad (Reddi et al., 2018), AdaBound (Luo et al., 2019), RAdam (Liu et al., 2020) and AdaBelief (Zhuang et al., 2020). The main difference is that the diagonal preconditioning in APOLLO is directly derived from the quasi-Newton updating formula (6). In terms of memory efficiency, Anil et al. (2019) and Chen et al. (2020) further reduces the memory cost of adaptive methods, and Agarwal et al. (2019) proposed efficient full-matrix adaptive regularization.

**Stochastic Second-Order Hessian-Free Methods.** Stochastic Second-Order Hessian-Free methods (Martens, 2010; Martens & Sutskever, 2011) implicitly solve quadratic models using matrix-vector products. Dauphin et al. (2014) argued the existence of saddle points and proposed a method to rapidly escape them. K-FAC (Martens & Grosse, 2015) computes a second-order step by constructing an invertible approximation of the Fisher information matrix in an online fashion. Shampoo (Gupta et al., 2018) approximates the Fisher information matrix using low-rank decomposition. Recently, Yao et al. (2020) proposed AdaHessian, which approximates the Hessian diagonal using Hutchinson's method. These second-order methods differ from APOLLO mainly in the request of second-order information of the objective function at each iteration.

## 6 CONCLUSION AND EXTENSIONS

We have introduced APOLLO, a simple and computationally efficient quasi-Newton algorithm for nonconvex stochastic optimization. This method is aimed towards large-scale optimization problems in the sense of large datasets and/or high-dimensional parameter spaces such as machine learning with deep neural networks. Experimental results on three CV and NLP tasks demonstrate the effectiveness of APOLLO, in terms of both convergence speed and generalization performance. In Appendix C, we briefly outline a few extensions to APOLLO that we want to explore in future work.

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
