# OpenReview forum: "Apollo: An Adaptive Parameter-wised Diagonal Quasi-Newton Method for Nonconvex Stochastic Optimization"
_ICLR.cc/2022/Conference — ICLR 2022 Submitted_

### Official Review · Reviewer_p4ji · 2021-10-22

**Correctness:** 3
**Technical Novelty And Significance:** 2
**Empirical Novelty And Significance:** 3
**Recommendation:** 6
**Confidence:** 3

**Main Review:**

*Strengths*: Overall, the paper is well-written and well-motivated,
the background material is sufficient, and the computational experiments
are comprehensive and significant. I also appreciate the efforts the
authors made to keep the presentation simple and concise.

*Weaknesses*: I have a few technical issues with the presentation
of the results:
1. Theorem 2 and Corollary 2.1 are not "convergence" results, but
rather regret bounds. For completeness, the authors should present
a true convergence result, like in Theorem 3, in which a relevant
residual, e.g. $\min_{t\in[T]} E [\|\nabla f(\theta_t)\|] $
or $\min_{t\in[T]}E[f(\theta_{t})-f_{*}]$, is proven to go to zero
as $T\to\infty$. It should also be expected that the produced convergence rate
be similar to other quasi-Newton convergence rates for the convex setting.

2. The convergence rate established in Theorem 3 in the nonconvex case
is on the same order of complexity as other Adam-based methods (in
fact, the proof of this result is based on prior analyses of Adam-type
methods). This result is not encouraging from a theoretical point
of view as there are other works in the literature (see, for example,
[1]) that make use of second-order information (through first-order
approximations) that have better convergence rates than purely first-order
methods. The authors should comment on why their method cannot achieve
such a level of acceleration, e.g., due to rectification or the use
of a weakened secant equation, and/or the advantage of their method
over other **accelerated** methods like in [1].

Aside from the above points, I only have a few minor issues with some
of the other material in the paper:

1. [p. 6] Give the precise definition of the regret $R_{T}$ (it is only defined in the Appendix).
2. *Minor typos*:
- [p. 5] All instances of "problem" should be plural in the
first paragraph of Subsection 3.3.
- [p. 6] Remove the period in equation (11).
- [p. 6] ... Similar to previous **works** ...

[1] Carmon, Y., Duchi, J. C., Hinder, O., & Sidford, A. (2018).
Accelerated methods for nonconvex optimization. *SIAM Journal on
Optimization, 28*(2), 1751-1772.

**Summary Of The Paper:**

This paper presents a first-order quasi-Newton method, named APOLLO,
for solving stochastic nonconvex finite sum problems with a large
number of data points. Each iteration of the method consists of computing
a sparse and positive definite (diagonal) approximation of the objective
function's Hessian, followed a quasi-Newton step. A regret bound is
established for the convex setting and a complexity bound is established
for the nonconvex setting, based on prior results on Adam-type optimizers.
Finally, a comprehensive set of numerical experiments on three common
tasks in vision and language is presented to show the superiority
of the proposed method.


**Summary Of The Review:**

This paper is overall well-written and contains a comprehensive and
significant set of numerical experiments. A few gaps in the presentation
of the theoretical contributions, however, prevent it from receiving
a stronger score.

---

> ### Author Response · Authors · 2021-11-22
> **Response to Reviewer p4ji**
>
> Thanks for your comments and positive feedback. We respond below to your questions and comments.
>
> > Theorem 2 and Corollary 2.1 are not "convergence" results, but rather regret bounds.
>
> We used regret bound in Theorem 2 and Corollary 2.1 to follow the research line of optimization for DNNs.
> We will improve this theoretical part of Apollo to provide a true convergence result.
>
> > The convergence rate established in Theorem 3 in the nonconvex case is on the same order of complexity as other Adam-based methods (in fact, the proof of this result is based on prior analyses of Adam-type methods). This result is not encouraging from a theoretical point of view as there are other works in the literature (see, for example, [1]) that make use of second-order information (through first-order approximations) that have better convergence rates than purely first-order methods. The authors should comment on why their method cannot achieve such a level of acceleration, e.g., due to rectification or the use of a weakened secant equation, and/or the advantage of their method over other accelerated methods like in [1].
>
> We appreciate your suggestion and will improve the theoretical convergence of Apollo.

---

### Official Review · Reviewer_qbeR · 2021-10-31

**Correctness:** 3
**Technical Novelty And Significance:** 2
**Empirical Novelty And Significance:** 2
**Recommendation:** 6
**Confidence:** 3

**Main Review:**

Strengths:
The authors provided theoretical guarantee of the proposed algorithm, Apollo, and showed good empirical performance of it.

Weaknesses:
- The algorithm imposed the restriction that parameters have to be updated component-wise; while this is applicable to neural-network type problems (i.e., each layer may be updated separately during training), I don't see how this can be easily applied to other ML problems.
- I'm not able to judge the technical correctness of the statements in Theorems 1,2,3 from the main paper; One question I have is about the parameter-wise update of the gradient/Hessian; this results in a different type of algorithm than SGD-variants; While I'm not a hundred percent sure, I believe this is also not the case for Adam/Ada-grad type algorithms. How does this not affect the convergence guarantee? Does this type of parameter-wise update fit the framework of generalized-Adam?

**Summary Of The Paper:**

The paper developed a new quasi-Newton algorithm for stochastic non-convex optimization. In contrast to existing works, it uses a (rectified/capped) diagonal matrix to approximate the Hessian, and incorporated techniques to reduce the stochastic variance. It shows first-order convergence guarantee of the algorithm and provided empirical evaluation on 2 CV and 2 NLP datasets.

**Summary Of The Review:**

The paper could be a good contribution to the literature (as well as for practice), however I have several concerns currently holding me back, which I hope the authors can address in the rebuttal (they are all related to the parameter-wise update regime enforced by the algorithm; please see my comments in the main review section)

---

> ### Author Response · Authors · 2021-11-22
> **Response to Reviewer qbeR**
>
> Thanks for your comments and positive feedback. We respond below to your questions and comments.
>
> > The algorithm imposed the restriction that parameters have to be updated component-wise; while this is applicable to neural-network type problems (i.e., each layer may be updated separately during training), I don't see how this can be easily applied to other ML problems.
>
> We concede that Apollo is indeed designed for optimizing deep neural networks. We think it is an interesting direction of future work to explore the scenarios and effectiveness of applying Apollo to non-DL problems.
>
> > I'm not able to judge the technical correctness of the statements in Theorems 1,2,3 from the main paper; One question I have is about the parameter-wise update of the gradient/Hessian; this results in a different type of algorithm than SGD-variants; While I'm not a hundred percent sure, I believe this is also not the case for Adam/Ada-grad type algorithms. How does this not affect the convergence guarantee? Does this type of parameter-wise update fit the framework of generalized-Adam?
>
> Apollo is different with SGD and Adam-type methods in several aspects. But Apollo still belongs to the generalized-Adam family because this family is quite general.
>
> > Following the point above, in the experiment section, for the other benchmarking algorithms, how are the network parameters updated? Did the authors only compare Apollo with other algorithms in the setup where the parameters are updated layer-wise? If yes, perhaps the authors should provide additional ablation study comparing layer-wise update with traditional back-propagation.
>
> For other optimization methods, including SGD, Adam and their variants, there is no difference if their parameters are updated in the layer-wise way. This is because in these methods, each dimension of parameters are updated independently. In Apollo, since we need to estimate the diagonal Hessian approximation, we need to consider the coordinations across different dimensions.

---

> > ### Comment · Reviewer_qbeR · 2021-11-24
> > **Re: authors' feedback**
> >
> > We thank the authors for your feedback on our review. We realized that the last part of our review (regarding "layer-wise" update) was a bit unwarranted and should be "component-wise" instead, and realized that since for SGD variants, the model parameters are not updated coordinate-wise, it is not possible to do a direct comparison. After reading the authors' feedback, our review score remains the same but we would not argue for an acceptance for the paper.

---

### Official Review · Reviewer_XwAE · 2021-11-02

**Correctness:** 2
**Technical Novelty And Significance:** 2
**Empirical Novelty And Significance:** 2
**Recommendation:** 3
**Confidence:** 4

**Main Review:**

Strengths:
There is certain novelty in the proposed method, and the issues of nonconvexity, computational efficiency, and stochastic variance that are addressed in this work are significant.

Weaknesses:
1.	In Theorem 2 and Corollary 2.1, there is an assumption on $\|\|D_{t-1}\|\|\_1$ and $\|\|D_t\|\|\_1$, but $D\_t$ is a variable in the iterative update. Similarly, Theorem 3 has the assumption $\frac{D_{t-1},j}{\eta_{t-1}} \leq \frac{D_{t,j}}{\eta_t}$. How strong are these assumptions? Can they be satisfied during the iterations?
2.	One of the key points in this work is that the parameter-wise update of $B_t$ trades off between secant condition and weak secant condition. Intuitively, one would expect that the number of decoupled parameters $L$ has influence on the performance both theoretically and numerically. However, $L$ does not appear in the convergence analysis or the experiments.
3.	In the experiments, it is claimed that the proposed method outperforms the compared first-order methods on convergence speed. However, it is only true in terms of epochs, but not true in terms of runtime. It is quite common that a quasi-second-order method converges faster in terms of iterations, but the real question is how efficient it is in runtime.
4.	It is suggested that the notation for the Lipchitz constant in Theorem 3 should be changed, since L already denotes the number of decoupled parameters.


**Summary Of The Paper:**

This work proposes a quasi-Newton method APOLLO for nonconvex stochastic optimization. Based on a parameter-wise weak secant condition, a diagonal approximation of the Hessian is constructed, and rectified absolute value is adopted on the approximation. A step-size bias correction technique is used to mitigate stochastic gradient variance. Theoretical convergence analysis is provided under a convex online setting and a nonconvex stochastic setting. Experiments on CV and NLP tasks demonstrate the effectiveness and stability of the proposed method.

**Summary Of The Review:**

Though there is certain novelty in this work, the theoretical contributions seem to be flawed, and the numerical results are not convincing, as explained in the main review. Thus, my current recommendation for this work is rejection.

---

> ### Author Response · Authors · 2021-11-22
> **Response to Reviewer XwAE**
>
> Thanks for your comments and feedback, especially those on theoretical parts. We will revise our paper to reflect these comments.

---

> ### Comment · Reviewer_XwAE · 2021-11-29
> **keep my score**
>
> I have read the response from the authors and the comments from other reviewers, and I would like to keep my score unchanged.

---

### Official Review · Reviewer_LyjG · 2021-11-03

**Correctness:** 3
**Technical Novelty And Significance:** 2
**Empirical Novelty And Significance:** 2
**Recommendation:** 5
**Confidence:** 4

**Main Review:**

The paper presents a diagonal approximation to the Hessian. The presentation is very clear, and the algorithm is simple and easy to implement. The experiments show improvements on several datasets.

Diagonal approximation of quasi-Newton methods has been studied before, as the authors point out. The main contribution of the paper is to combine it with several other ideas, such as rectification (also done by AdaHessian), momentum via moving averages, learning rate warmup and step size bias correction.

The sentence in the introduction "these adaptive optimization methods may converge to bad/suspicious local optima" is rather strange, given the widespread use of these methods in industrial and academic settings. Seems unfair to castigate methods that have been used in so many applications on the basis of one isolated study. Adaptive methods are the methods of choice for NLP and CV tasks for several years, notwithstanding the claim made in the first para of page 2 (those papers are from several years ago).

The claim that Apollo "significantly outperforms SGD and variants of Adam, in terms of both convergence speed and generalization performance" is not supported by the results. The image classification results use non-standard architectures, and the improvement in accuracy is small. The appendix shows that Apollo is 20-50% slower than SGD and Adam --- it is hard to tell from the graphs, but it is quite likely that in the same time, SGD and Adam could have produced the same accuracy. Please include a test accuracy vs time plot in the appendix.

Furthermore, given that the current method is estimating the Hessian, it should be compared with the second order methods such as  Shampoo (Anil et al), as these methods are able to outperform SGD and Adam on large problems.

**Summary Of The Paper:**

The paper presents a diagonal quasi-Newton method, by approximating the Hessian with a diagonal matrix. The paper proves a regret bound for the method in the convex case, and shows that in the non-convex setting the expected norm of the gradients goes to 0. The paper then provides experimental results on two image datasets, a small translation task and a small language modeling task.



**Summary Of The Review:**

The paper presents an optimizer by combining several ideas from previous papers, and shows small improvements on some datasets.

---

> ### Author Response · Authors · 2021-11-22
> **Response to Reviewer LyjG**
>
> Thanks for your comments and positive feedback, especially the one to compare with second order methods such as Shampoo and to include figures of test accuracy vs time plot. We will revise our paper to reflect these comments.

---

### Decision · Program_Chairs · 2022-01-20

**Decision:**

Reject

**Comment:**

This paper proposes a diagonal approximation to the Hessian in a quasi-Newton method for non convex stochastic optimization problems. They combine several good existing ideas and show empirically that the method performs well on several learning tasks, but reviewers found that the comparisons were limited in that as an (approximate) second order method, it would be more fair to compare to other second-order methods rather than largely focusing on SGD and some variants of ADAM. Overall, reviewers found the novelty limited and had some concerns about the strength of assumptions, parameter-wise updates, and some more minor comments on gaps in the presentation. The author response did not fully convince the borderline/negative reviewers, though the paper includes good ideas that would potentially be well received in a future revision.